

# Prevalence and correlates of psychological distress among diabetes mellitus adults in the Jilin province in China: a cross-sectional study

Shuang Qiu[1,*], Hongxuan Sun[1,*], Yawen Liu[1], Joseph Sam Kanu[1], Ri Li[1], Yaqin Yu[1], Xufeng Huang[2], Bo Li[1] and Xiangyang Zhang[3]

[1] Department of Epidemiology and Biostatistics, School of Public Health, Jilin University, Changchun, China
[2] Illawarra Health and Medical Research Institute, School of Medicine, University of Wollongong, Wollongong, Australia
[3] Psychiatry Research Center, Beijing Hui-Long-Guan Hospital, Peking University, Beijing, China
* These authors contributed equally to this work.

## ABSTRACT

**Background.** Psychological disorders are common in diabetes mellitus (DM) patients, and the aim of this study was to estimate the prevalence of psychological distress and to determine the influence factors associated with psychological distress among DM patients in the Jilin province of China.

**Methods and Materials.** Multistage, stratified cluster sampling was used in this cross-sectional study. The 12-item General Health Questionnaire (GHQ-12) was used to assess psychological status with the total score of $\geq 4$ as the threshold for psychological distress.

**Results.** A total of 1,956 subjects with DM were included in the study. Out of this total diabetic participants, 524 (26.8%) had psychological distress. Multiple logistic regression analysis showed that low educational level, divorce or separation from one's spouse, low family average monthly income, short sleep duration, being aware of DM status, and multiple co-morbidities are positively associated with psychological distress (all $P < 0.05$).

**Conclusions.** This study revealed a high rate of psychological distress among DM population in Jilin province. Low educational level, divorce or separation from one's spouse, low family average monthly income, short sleep duration, awareness of DM status, and multiple co-morbidities are all associated with psychological distress among our study subjects. Interventions to control these factors are needed to address the psychological problems among diabetics in Jilin Province.

## INTRODUCTION

Diabetes mellitus (DM) is one of the most common chronic metabolic diseases, and is a major health problem in many countries (*Fisher et al., 2012*; *Whiting et al., 2011*). In 2010, estimated 285 million people were affected by DM globally, and DM prevalence is

Corresponding authors
Bo Li, li_bo@jlu.edu.cn
Xiangyang Zhang,
xiang.y.zhang@uth.tmc.edu

expected to increase to 438 million people by 2030 (*Raval et al., 2010*). Rapid economic development and the subsequent changes in lifestyle have led to the widespread occurrence of DM (*Guariguata et al., 2014*). China has the largest diabetic population worldwide, and DM prevalence is estimated to be 11.6%, representing nearly 113.9 million Chinese adults (*Xu et al., 2013*).

DM is not only associated with micro and macro vascular complications, such as increased risk of cardiovascular disease, diabetic nephropathy, retinopathy, neuropathy and lower extremity amputations, but may also influence physical functions, social interaction and mental well-being (*Jhita et al., 2014*; *Shin et al., 2014*).

Accumulating evidence suggests that psychological disorders are common in DM patients, and that DM patients are at substantial risk of heightened depression, anxiety, and stress (*Anderson et al., 2001*; *Atlantis et al., 2012*; *Raymond & Lovell, 2016*; *Roy & Lloyd, 2012*; *Yu et al., 2016*). Psychological disorders of DM patients are significantly associated with non-adherence to healthy lifestyle guidelines, thus increasing the risk for serious complications that may decrease patients' quality of life and result in premature death (*Ducat, Philipson & Anderson, 2014*; *Gonzalez et al., 2008*). Using the 12-item General Health Questionnaire (GHQ-12), *Zubair, Mansoor & Rana (2014)* reported that 68.5% of their study participants that scored more than 4 points had depressive symptoms, indicating that DM patients with psychological distress are more prone to depression. Thus, GHQ-12 as a general measure of psychological distress is a simple and useful instrument for healthcare practitioners to evaluate the overall psychological status of DM patients.

Although many studies have focused on the link between psychological disorders and DM, epidemiological data on the prevalence and correlates of psychological distress among DM population in Jilin Province are limited. The aims of the study were: (1) to estimate the prevalence of psychological distress among DM patients in Jilin Province; (2) to determine the influence factors associated with psychological distress among DM patients in Jilin Province; and (3) to provide proper references for relevant departments.

## MATERIALS AND METHODS

### Study population

This study was conducted using a survey of chronic disease and risk factors among adults (aged 18–79 years) in the Jilin province of China in 2012. The sample size was calculated considering the prevalence of chronic diseases investigated in this survey (varying from 0.02 to 0.99), two-tailed confidence level ($\alpha$) of 95%, and absolute error varying from 0.01 to 0.10 were used. The ultimate target sample size was estimated to be 25,240, which accounts for approximately 1‰ of the total adult population of Jilin province. We used multistage stratified cluster random sampling method to select a representative sample of permanent residents who had lived in Jilin province within nine regions (Changchun, Jilin, Siping, Liaoyuan, Tonghua, Baishan, Songyuan, Baicheng and Yanbian) for at least six months. Then, we randomly selected clusters of four districts or counties from each of the nine regions using probability proportional to size (PPS) sampling. We selected 32 districts

or counties, 95 towns or communities, and 45 units. Finally, we randomly selected one adult resident from each household in the selected units. The detailed sampling process has been described previously (*Wang et al., 2014*; *Wang et al., 2015*).

All participants gave informed consents before being recruited in the study. We adhered to the bioethics principles of the Declaration of Helsinki, and our study was authorized by the Ethics Committee of the School of Public Health of Jilin University (Reference Number: 2012-R-011) and the Bureau of Public Health of Jilin Province (Reference Number: 2012-10).

## Data collection and measurement
### Personal health questionnaire

We used an interviewer administered questionnaire to obtain information from our study subjects. The questionnaire was designed by our project team, and primarily included demographic factors (gender, age, race, community, educational level, marital status, occupation, and family average monthly income), as well as other information such as daily sleep duration and anthropometric measurements. The Mandarin Chinese language was used to design the questionnaire, and the questionnaire was administered in Chinese by investigators who could speak Mandarin Chinese and had received uniform training on the administration of the questionnaire.

### GHQ-12

The GHQ-12, a shortened version of a 60-item screening tool developed by Goldberg in 1970, is a widely-used instrument for assessing psychiatric morbidity, especially to screen for depression and anxiety disorders (*Goldberg et al., 1997*). It has recently been adapted as a screening tool for psychological distress in various populations (*Cuellar-Flores et al., 2014*; *Higuchi et al., 2016*; *Yi et al., 2016*). It has been well-validated for general Chinese populations (*Yang, Huang & Wu, 2003*; *Zhou et al., 2013*). The GHQ-12 includes 12 questions, each with two response options (Yes or No). Each item is scored as either 0 (less or no more than usual) or 1 (more or much more than usual), for a maximum total score of 12. Higher scores indicate more psychological problems. According to the total score of the GHQ-12, participants scoring 4 or more are considered to be in psychological distress (*Phillips et al., 2009*).

### Baseline clinical data

Fasting plasma glucose (FPG) was measured from participants who had fasted for 10 or more hours overnight. A small drop of blood was taken from one finger, placed onto a strip of paper and the glucose level measured using a Bayer Bai Ankang fingertip blood glucose monitoring machine.

DM was defined as either a fasting plasma glucose level $\geq 7.0$ mmol/L, according to the Chinese Guidelines on the prevention and treatment of Type 2 Diabetes (*Society, 2008*), or self-reported diagnosed DM by a medical doctor in a hospital. Those self reported patients were considered to be aware of their DM status, and those who were only diagnosed during our investigation were considered to be unaware of their DM status.

Body mass index (BMI) was categorized into two groups: underweight or normal weight (BMI < 24 kg/m$^2$), and overweight or obesity (BMI ≥ 24 kg/m$^2$) (*Zhou, 2002*).

Number of co-morbid diseases was defined as a count of the following chronic diseases reported simultaneously with DM by patients, including hyperlipidemia, hypertension, cerebrovascular disease, coronary heart disease and myocardial infarction, cataract, gout, and hyperthyroidism. Chronic diseases were determined by participants who self-reported having previous diagnosis of the diseases from a health professional. DM patients were categorized into four groups: without any co-morbid disease, only one kind of co-morbid disease, two, three or more co-morbid diseases.

### Sociodemographics

Community was divided into two groups: urban (including those who lived in cities or towns) and rural (including the rest of the entire population).

We divided education into three levels: primary school and below (including those who never attended school and those whose highest level of education was primary school), junior high school (including those whose highest level of education was junior high school), and senior high/technical secondary school and above (including those whose highest level of education was high school, technical secondary school, or university degree).

Marital status was divided into four groups: married (including those who were married and those co-habiting), never married, divorced or separated, and widowed.

We divided occupation into three groups: manual workers (including farmers, service workers, and those who were engaged in production work), mental workers (including office and other technical staff), and others (including students, unemployed, and retirees) (*Wang et al., 2014*; *Zhang et al., 2016*).

Family average monthly income was calculated as the ratio of the total family income to the number of family members in the household, and was classified into five groups: low income group (<500 yuan/month), lower-middle group (500–1,000 yuan/month), middle group (1,000–2,000 yuan/month), higher-middle group (2,000–3,000 yuan/month), and high income group (≥3,000 yuan/month) in line with the classification of the Jilin Provincial Bureau of Statistics.

## Quality control

We performed a pilot study to assimilate the main study using a total of 190 participants. Issues that aroused during the pilot study were addressed before the main study. All data were processed by parallel double entry using EpiData 3.1 database.

## Statistical analysis

Descriptive statistics were used to summarize the numbers and percentages for categorical variables, and mean (standard deviation—S.D) and median (IQR) for continuous variables. Univariate logistic regression analyses were used to compare the rates of psychological distress between different participants' characteristics (gender, age, race, community, educational level, marital status, occupation, family average monthly income, daily sleep duration, BMI, family history of DM, number of co-morbid diseases, and the awareness of DM). Variables that were significant in the univariate logistic analyses were included

in the final multiple logistic regression analyses. Multiple logistic regression analyses were performed to explore the association between participants' characteristics and psychological distress. The statistical tests were 2-tailed, and significance was defined as $P < 0.05$. All data were analyzed using SPSS (Version 21.0, IBM SPSS, IBM Corp, Armonk, NY, USA).

## RESULTS

A total of 1,956 DM participants were recruited in this study with a mean age of 55.68 years (S.D. = 10.26). We present participants demographic characteristics and other information in Table 1. The proportions of males (49.7%) and females (50.3%) were almost the same, as well as the residence location of the participants—urban (50.8%) or rural (49.2%). Nine hundred and fifty-nine participants (49%) were 45–59 years old, which was the largest age group. More than 90% of participants were Han nationality. We also observed that 60.5% of participants attained a junior high school or higher level of education; 89.4% of participants were married; 56.3% of participants had sleep duration of ≥7 h per day; 30.7% of participants had family average monthly income of 1000–2000 yuan; 71.5% of participants had no history of DM; and 67.6% of participants were overweight or obese. More participants were engaged in manual work (47.3%) than mental labor (14.0%), with 68.9% of participants being aware of their DM status. Less than half (47.0%) of participants had no other diseases.

We further performed analysis to explore the factors associated with psychological distress of DM participants. Population characteristics according to psychological distress are shown in Table 2. Mean GHQ-12 score was 2 (IQR 0-4). Out of the 1,956 patients, 524 (26.8%) participants had psychological distress. Univariate analysis revealed that the rate of psychological distress significantly differed by gender, community, educational level, marital status, occupation, family average monthly income, daily sleep duration, number of co-morbid diseases, and the awareness of DM (all $P < 0.05$), but not by age ($P = 0.056$), race ($P = 0.472$), BMI ($P = 0.073$), and family history of DM ($P = 0.126$). Compared with male, female were more likely to have psychological distress (OR = 2.046 [1.666, 2.513]). Urban participants were less prone to psychological distress (OR = 0.628 [0.513, 0.769]), compared with rural participants. Participants whose highest level of education was primary school were associated with a higher prevalence of psychological distress than those whose highest level of education was senior high school/technical secondary school (OR = 2.472 [1.938, 3.154]). The *ORs* (95% CI) were 3.074 [1.651, 5.725] for divorce /separate DM participants, and 1.692 [1.183, 2.420] for widowed DM participants. Compared with manual workers, mental workers were less likely to be distressed (OR = 0.533 [0.380, 0.747]). The prevalence of psychological distress was higher among participants whose family average monthly income was less than 500 yuan (OR = 3.100 [1.958, 4.910]), or 500–1,000 yuan (OR = 2.214 [1.387, 3.535]) than those whose income was ≥3,000 yuan. Participants with sleep duration of <7 h per day were associated with a higher prevalence of psychological distress than participants with sleep duration of ≥7 h (OR = 1.464 [1.198, 1.790]). Participants whose co-morbid disease was 2 (OR = 1.587 [1.194, 2.110]), and ≥3 (OR = 2.993 [2.033, 4.407]) were more prone to have higher rate of psychological distress than those without any other diseases. Participants who were aware of their DM status

**Table 1  Participants characteristics ($N = 1,956$).**

| Characteristics | n | % |
|---|---|---|
| **Gender** | | |
| Male | 972 | 49.7 |
| Female | 984 | 50.3 |
| **Age group(years)** | | |
| 18–44 | 267 | 13.7 |
| 45–59 | 959 | 49.0 |
| 60–79 | 730 | 37.3 |
| **Race** | | |
| Han nationality | 1,795 | 91.8 |
| Non-Han minority | 161 | 8.2 |
| **Community** | | |
| Urban | 994 | 50.8 |
| Rural | 962 | 49.2 |
| **Educational level** | | |
| Primary school and below | 773 | 39.5 |
| Junior high school | 520 | 26.6 |
| Senior high/technical secondary school and above | 663 | 33.9 |
| **Marital status** | | |
| Married | 1,748 | 89.4 |
| Never married | 25 | 1.3 |
| Divorce or separate | 41 | 2.1 |
| Widowed | 142 | 7.3 |
| **Occupation** | | |
| Manual worker | 925 | 47.3 |
| Mental worker | 274 | 14.0 |
| Others | 757 | 38.7 |
| **Family average monthly income (yuan/month)** | | |
| <500 | 487 | 24.9 |
| 500–1,000 | 397 | 20.3 |
| 1,000–2,000 | 600 | 30.7 |
| 2,000–3,000 | 320 | 16.4 |
| ≥3,000 | 152 | 7.8 |
| **Daily sleep duration (hours)** | | |
| <7 | 854 | 43.7 |
| ≥7 | 1,102 | 56.3 |
| **BMI[a] (kg/m$^2$)** | | |
| <24 | 601 | 30.7 |
| ≥24 | 1,323 | 67.6 |
| **Family history of DM** | | |
| No | 1,398 | 71.5 |
| Yes | 558 | 28.5 |

**Table 1** (*continued*)

| Characteristics | n | % |
|---|---|---|
| **Number of co-morbid disease** | | |
| 0 | 920 | 47.0 |
| 1 | 601 | 30.7 |
| 2 | 312 | 16.0 |
| ≥3 | 123 | 6.3 |
| **Awareness of DM** | | |
| Yes | 1,347 | 68.9 |
| No | 609 | 31.1 |

**Notes.**

$^a$Total percentage did not equal to 100% because of missing data.

were more likely to have psychological distress than those who were unaware (OR = 1.523 [1.214, 1.911]).

Table 3 shows the results of multivariate logistic regression analysis. This analysis revealed no significant difference in psychological distress by community and occupations ($P > 0.05$). However, we found significant differences in psychological distress by educational level, marital status, family average monthly income, daily sleep duration, the number of co-morbid diseases, and the awareness of DM after considering the effect of confounding factors ($P < 0.05$).

Participants who never attended school and those whose highest level of education was primary school were associated with a higher prevalence of psychological distress than those whose highest level of education was senior high school/technical secondary school (OR = 1.474 [1.087, 1.999], $P = 0.013$). Divorced or separated participants were more likely to have psychological distress than married participants (OR = 3.434 [1.772, 6.653], $P < 0.001$). Participants whose family average monthly income was less than 500 yuan (OR = 2.282 [1.377, 3.781], $P = 0.01$), or 500–1,000 yuan (OR = 1.728 [1.045, 2.856], $P = 0.033$) were associated with a higher prevalence of psychological distress than those whose income was ≥3,000 yuan. Participants who were aware of their DM status were more likely to develop psychological distress (OR = 1.305 [1.022, 1.667], $P = 0.033$). Participants with sleep duration of <7 h per day were more likely to have psychological distress than participants with sleep duration of ≥7 h (OR = 1.522 [1.231, 1.883], $P < 0.001$). Participants who had two kinds of other diseases (OR = 1.534 [1.129 2.084], $P = 0.006$), and three or more other diseases (OR = 2.473 [1.635 3.739], $P < 0.001$) were associated with a lower prevalence of psychological distress than participants without any other diseases.

## DISCUSSION

This population-based study adds to the literature that the prevalence of psychological distress (GHQ-12 total score ≥4) among DM patients in the Jilin province of China is 26.8%, higher than that the 18.9% reported for England and Scottish DM populations (*Hamer et al., 2010*), and also higher than the Quebec and Canada DM populations (22.1%) (*Smith et al., 2013*). The inconsistency may be partly due to the disparities in assessment instruments, sampling methods, and the definition of psychological distress.

**Table 2 Univariate analysis of factors associated with psychological distress N = 1,956.**

| Characteristics | Non-distressing | Distressing | P | OR | 95% CI |
|---|---|---|---|---|---|
| **Gender** | | | <0.001 | | |
| Male | 779(80.1) | 193(19.9) | | 1.000 | |
| Female | 653(66.4) | 331(33.6) | | 2.046 | 1.666–2.513 |
| **Age group(years)** | | | 0.056 | | |
| 18–44 | 195(73.0) | 72(27.0) | | 1.000 | |
| 45–59 | 724(75.5) | 235(24.5) | | 0.879 | 0.646–1.196 |
| 60–79 | 513(70.3) | 217(29.7) | | 1.146 | 0.837–1.567 |
| **Race** | | | 0.472 | | |
| Han nationality | 1,311(73.0) | 485(27.0) | | 1.000 | |
| Non-Han minority | 121(75.6) | 39(24.4) | | 0.871 | 0.598–1.268 |
| **Community** | | | <0.001 | | |
| Rural | 660(68.6) | 302(31.4) | | 1.000 | |
| Urban | 772(77.7) | 222(22.3) | | 0.628 | 0.513–0.769 |
| **Educational level** | | | <0.001 | | |
| Senior school and above[a] | 538(81.1) | 125(18.9) | | 1.000 | |
| Junior high school | 403(77.5) | 117(22.5) | | 1.250 | 0.941–1.659 |
| Primary school and below | 491(63.5) | 282(36.5) | | 2.472 | 1.938–3.154 |
| **Marital status** | | | <0.001 | | |
| Married | 1,303(74.5) | 445(25.5) | | 1.000 | |
| Never married | 19(76.0) | 6(24.0) | | 0.925 | 0.367–2.330 |
| Divorce or separate | 20(48.8) | 21(51.2) | | 3.074 | 1.651–5.725 |
| Widowed | 90(63.4) | 52(36.6) | | 1.692 | 1.183–2.420 |
| **Occupation** | | | 0.001 | | |
| Manual worker | 652(70.5) | 273(29.5) | | 1.000 | |
| Mental worker | 224(81.8) | 50(18.2) | | 0.533 | 0.380–0.747 |
| Others | 556(73.4) | 201(26.6) | | 0.863 | 0.697–1.070 |
| **Monthly income[b]** | | | <0.001 | | |
| ≥3,000 | 126(82.9) | 26(17.1) | | 1.000 | |
| 2,000–3,000 | 213(83.9) | 41(16.1) | | 0.933 | 0.544–1.598 |
| 1,000–2,000 | 494(79.3) | 129(20.7) | | 1.265 | 0.795–2.014 |
| 500–1,000 | 302(68.6) | 138(31.4) | | 2.214 | 1.387–3.535 |
| <500 | 297(61.0) | 190(39.0) | | 3.100 | 1.958–4.910 |
| **Daily sleep duration (hours)** | | | <0.001 | | |
| ≥7 | 843(76.5) | 259(23.5) | | 1.000 | |
| <7 | 589(69.0) | 265(31.0) | | 1.464 | 1.198–1.790 |
| **BMI* (kg/m²)** | | | 0.073 | | |
| <24 | 424(70.5) | 177(29.5) | | 1.000 | |
| ≥24 | 985(74.5) | 338(25.5) | | 0.822 | 0.663–1.019 |
| **Family history of DM** | | | 0.126 | | |
| No | 1,037(74.2) | 361(25.8) | | 1.000 | |
| Yes | 395(70.8) | 163(29.2) | | 1.185 | 0.953–1.475 |

**Table 2** (*continued*)

| Characteristics | Non-distressing | Distressing | P | OR | 95% CI |
|---|---|---|---|---|---|
| **Number of co-morbid disease** | | | <0.001 | | |
| 0 | 714(77.6) | 206(22.4) | | 1.000 | |
| 1 | 438(72.9) | 163(27.1) | | 1.290 | 1.017–1.636 |
| 2 | 214(68.6) | 98(31.4) | | 1.587 | 1.194–2.110 |
| ≥3 | 66(53.7) | 57(46.3) | | 2.993 | 2.033–4.407 |
| **Awareness of DM** | | | <0.001 | | |
| No | 479 (78.7) | 130 (21.3) | | 1.000 | |
| Yes | 953 (70.7) | 394 (29.3) | | 1.523 | 1.214–1.911 |

**Notes.**
*Total percentage did not equal to 100% because of missing data. Data are presented as *n*(%); OR, Odds Ratio; CI, confidence interval.
[a] Senior high/technical secondary school and above.
[b] Family average monthly income (yuan/month).

Similar to other studies (*Tellez-Zenteno & Cardiel, 2002*; *Yang, Li & Zheng, 2009*), we found that low educational levels were associated with increased risk of psychological distress. Patients with lower education levels were less knowledgeable about the association between DM and mental disorders, more difficult to understand how to moderate their mental stress, and consequently had a higher psychological well-being.

Many researchers concluded that marriage has a protective effect on mental health (*Fu & Noguchi, 2016*; *Hughes & Waite, 2009*). In our study, divorce or separation from one's spouse is the most significant factor associated with psychological distress. This is concordant with the findings of other researchers revealing that divorce or separation is associated with poor psychological well-being (*Tellez-Zenteno & Cardiel, 2002*). Divorced or separated participants may have issues of taking care of their parents, raising children alone, and trying to cope with the divorce and the burden of their disease. All these factors may have long-term negative effects on the mental health of these patients. Our results also showed no significant difference in psychological distress between widowed participants and married participants, which is inconsistent with the report by Akena and team (*Akena et al., 2014*). This may be related to the differences in widowhood duration and gender. Perkins and co-authors concluded that long-term (widowed 0 to 4 or 10+ years) widowhood predicts worse health for women, but not for men (*Perkins et al., 2016*). In our study, we adjusted the confounding influence of gender using multiple logistic regression, but we did not obtained information about widowhood duration which need further research.

Our study demonstrated that participants with low family average monthly income were more prone to psychological distress. This accords with a study in central China by Yang and team (*Yang, Li & Zheng, 2009*). Management of DM is associated with long term financial burden (*Arnold et al., 2016*). Failure to cope with this heavy financial burden might be associated with psychological disturbances. It is not surprising in our study that participants with low family average monthly income had a great likelihood of developing psychological distress. In a study among low and middle income countries, Leone et al.

**Table 3  Multivariate analysis of factors associated with psychological distress.**

| Variables | β | S.E. | Wald | P | OR | 95% CI Lower | 95% CI Upper |
|---|---|---|---|---|---|---|---|
| **Educational level** | | | | | | | |
| Senior high school and above[a] | | | | | 1.000 | | |
| Junior high school | −0.008 | 0.156 | 0.003 | 0.958 | 0.992 | 0.731 | 1.346 |
| Primary school and below | 0.388 | 0.156 | 6.217 | 0.013 | 1.474 | 1.087 | 1.999 |
| **Marital status** | | | | | | | |
| Married | | | | | 1.000 | | |
| Never married | 0.178 | 0.496 | 0.129 | 0.720 | 1.194 | 0.452 | 3.155 |
| Divorce or separate | 1.234 | 0.337 | 13.365 | <0.001 | 3.434 | 1.772 | 6.653 |
| Widowed | 0.132 | 0.199 | 0.441 | 0.507 | 1.141 | 0.773 | 1.686 |
| **Monthly income[b]** | | | | | | | |
| ≥3,000 | | | | | 1.000 | | |
| 2,000–3,000 | −0.112 | 0.282 | 0.157 | 0.692 | 0.894 | 0.515 | 1.554 |
| 1,000–2,000 | 0.086 | 0.248 | 0.120 | 0.729 | 1.090 | 0.671 | 1.771 |
| 500–1,000 | 0.547 | 0.256 | 4.545 | 0.033 | 1.728 | 1.045 | 2.856 |
| <500 | 0.825 | 0.258 | 10.259 | 0.001 | 2.282 | 1.377 | 3.781 |
| **Awareness of DM** | | | | | | | |
| No | | | | | 1.000 | | |
| Yes | 0.266 | 0.125 | 4.540 | 0.033 | 1.305 | 1.022 | 1.667 |
| **Daily sleep duration (hours)** | | | | | | | |
| ≥7 | | | | | 1.000 | | |
| <7 | 0.420 | 0.108 | 15.030 | <0.001 | 1.522 | 1.231 | 1.883 |
| **Number of co-morbid disease** | | | | | | | |
| 0 | | | | | 1.000 | | |
| 1 | 0.152 | 0.129 | 1.397 | 0.237 | 1.164 | 0.905 | 1.498 |
| 2 | 0.428 | 0.156 | 7.488 | 0.006 | 1.534 | 1.129 | 2.084 |
| ≥3 | 0.905 | 0.211 | 18.416 | <0.001 | 2.473 | 1.635 | 3.739 |

**Notes.**

Adjusted age and gender, Methord: Backward: Conditional, β represents the logistic regression coefficient, S.E. represents the standard error, OR, Odds Ratio; CI, confidence interval.

[a] Senior high/technical secondary school and above.

[b] Family average monthly income (yuan/month).

noted that the occurrence of depression among diabetic patients seems to be associated with lower socioeconomic status (*Leone et al., 2012*).

Participants who were aware of their DM status were more likely to have psychological distress. A meta-analysis of 13 studies by Nouwen and colleagues concluded that patients who were aware of DM had increased risk of depression relative to patients that were unaware (*Nouwen et al., 2011*). Our finding is consistent with the findings of other studies (*Domingo et al., 2015*; *Knol et al., 2007*; *Olvera et al., 2016*). These psychological problems in people with DM might be related to the consequences of the burden of DM. Participants who were aware of their DM status have to change their poor health behaviors (e.g., smoking, high fat diet, and low physical activity), or take their anti-diabetic medications.

The need for monitoring their blood glucose and medical complications or related diseases may induce psychological problems.

Poor sleep is an important etiology of depression (*Chang et al., 2012*; *Paunio et al., 2014*). This observation is in agreement with our finding that participants with short sleep durations had higher likelihood of developing psychological distress. Yu and team investigated the prevalence of depression among rural residents with DM patients in China, and reported that shorter sleep duration increases the risk of depressive symptoms (*Yu et al., 2016*). In addition, frequent urination that is a major symptom of DM may cause discomfort to diabetic patients and reduce their quality of sleep. Sleep quality can be a potential mediator between psychological distress and diabetes quality of life. In a study on veterans, Seligowski and co-authors found that sleep quality has a partial indirect effect on the relationships between the symptoms of depression and the quality of life of diabetic patients, and between the symptoms of anxiety and diabetes quality of life (*Seligowski et al., 2013*).

Participants who co-morbid more than or equal to two kinds of other diseases were associated with a higher prevalence of psychological distress. Similarly, a study from Palestine by *Sweileh et al. (2014)* showed that DM participants with combined multiple additional illnesses were more prone to develop depressive symptoms. Chronic pain, somatic discomfort, and restrictions on one's social life are common consequences of chronic medical illnesses which lower quality of life and increase the risk for psychological disorders (*Agborsangaya et al., 2013*).

Our study involved a large and representative sample of DM participants and a good response rate from these participants in Jilin province. We considered not only demographic factors but also baseline clinical factors possibly associated with psychological distress. Our study revealed a high rate of psychological distress among DM population in the Jilin province in China and some correlates of psychological distress. Prevention and treatment are urgently needed to address the public health problem of psychological distress among DM population and to prevent them from developing more serious psychological disorders. The Chinese government should pay attention to DM-related mental diseases, increase investment for health education and promotion, inform patients about the harm of DM-related mental diseases and formulate methods to reduce psychological distress, strengthen the primary mental health care system, increase financial subsidies for low income families, and put in place DM and other chronic diseases prevention and treatment measures.

However, our results may be biased by the method used to confirm DM. We only used FPG levels to confirm DM, as we did not perform postprandial glucose tolerance test or measure glycosylated hemoglobin levels. In addition, other unmeasured variables such as time since diagnosis, social support, and diabetes self-efficacy, might have been important to adjust for in our study. Third, because of the cross-sectional design, the interpretation of causal relationships between risk factors and the development of psychological distress is limited. Longitudinal studies are also needed to investigate the predictors of psychological distress in the Chinese DM population in the future.

## CONCLUSIONS

Psychological distress is highly prevalent among DM population in Jilin province, China. Several factors are associated with psychological distress among diabetic patients in Jilin Province (low educational level, divorce or separation from one's spouse, low family average monthly income, short sleep duration, being aware of DM status, and multiple co-morbidities). In order to effectively improve the quality of life of diabetics in this part of China, relevant health authorities should adequately address these factors.

## ACKNOWLEDGEMENTS

The authors would like to thank our survey team members (Changgui Kou, Yong Li, Qing Zhen, Yuchun Tao, Huan He, Yulu Gu and Chang Wang et al.), participants and volunteers for their effort in this study.

### Funding

This work was supported by the Scientific Research Foundation of the Health Bureau of Jilin Province, China (grant number: 2011Z16). The funders had no role in study design, data collection and analysis, decision to publish, or preparation of the manuscript.

### Grant Disclosures

The following grant information was disclosed by the authors:
Scientific Research Foundation of the Health Bureau of Jilin Province, China: 2011Z16.

### Competing Interests

The authors declare there are no competing interests.

### Author Contributions

- Shuang Qiu conceived and designed the experiments, performed the experiments, analyzed the data, contributed reagents/materials/analysis tools, wrote the paper, prepared figures and/or tables, reviewed drafts of the paper.
- Hongxuan Sun conceived and designed the experiments, performed the experiments, analyzed the data, contributed reagents/materials/analysis tools, prepared figures and/or tables, reviewed drafts of the paper.
- Yawen Liu and Yaqin Yu conceived and designed the experiments, performed the experiments, reviewed drafts of the paper.
- Joseph Sam Kanu performed the experiments, reviewed drafts of the paper.
- Ri Li analyzed the data, contributed reagents/materials/analysis tools, reviewed drafts of the paper.
- Xufeng Huang and Xiangyang Zhang reviewed drafts of the paper.
- Bo Li conceived and designed the experiments, performed the experiments, contributed reagents/materials/analysis tools, reviewed drafts of the paper.

## Human Ethics

The following information was supplied relating to ethical approvals (i.e., approving body and any reference numbers):

All participants gave their informed consent for inclusion before they participated in the study. We adhered to the bioethics principles of the Declaration of Helsinki, and our study was authorized by the Ethics Committee of Jilin University School of Public Health (Reference Number: 2012-R-011) and the Bureau of Public Health of Jilin Province (Reference Number: 2012-10).

## Data Availability

The raw data has been supplied as a Supplementary File.

## Supplemental Information

Supplemental information for this article can be found online at http://dx.doi.org/10.7717/peerj.2869#supplemental-information.

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
