# Peer review of "Prevalence and correlates of psychological distress among diabetes mellitus adults in the Jilin province in China: a cross-sectional study"

_PeerJ, doi:10.7717/peerj.2869_

## Round 0.1 · original submission · Major Revisions

Please consider the issues raised by the three reviewers, with particular attention to Methods, Statistics, and Discussion.

Reviewer 1 ·

Basic reporting

There is a clear, unambiguous, professional english language used throughout.
The Introduction and background are able to show context.
The literatures are well referenced & relevant.
The structure conforms to the journal standard,
The figures are relevant, high quality, well labelled & described.

Experimental design

-

Validity of the findings

There is room to improve

Annotated reviews are not available for download in order to protect the identity of reviewers who chose to remain anonymous.

·

Basic reporting

No comments

Experimental design

Materials and Methods

1. Page 3 – Lines 132-133: Could the authors briefly mention how was sample size calculated, instead of referencing them? Authors cited a reference to a study regarding the sampling technique. However, it is difficult to gauge if this study was part of the previous study (Wang et al., 2014) as cited. In addition, the sample participated in the study and the reported mean age of the respondents should be described in the results part, under descriptive statistics.
2. Page 4 - Line 146: The phrase “face-to-face interviews” is inappropriate for a quantitative study that uses a questionnaire. Please rephrase to “interviewer administered questionnaire”. And, in what language was the questionnaire administered? Was it in local language? If yes, did the questionnaire undergo forward and backward translations? Were psychometric properties assessed either by Exploratory Factor Analysis and further validated by Confirmatory Factor Analysis for this study population? Otherwise, please mention the Cronbach’s alpha values.
3. Page 4 – Please add a sub-topic, “baseline clinical data” under data collection and measurement. These measurements should be included under this sub-topic: fasting plasma glucose, and definitions – DM, BMI, other chronic illnesses, family history of DM. They are currently seemed to be a little out of space. The definitions for education level, marital status, occupation could be defined under “socio-demographics.” Please add other demographics variables as well in this section, how were they defined, coded and categorized?
4. Page 5 – Line 178: Classification of education level states primary school and “blow”. It should be “below.” Please correct the typo.
5. Page 5 – Lines 187-189: Could the authors cite with reference how was the classification for occupations obtained? The classified workers to groups seemed to be similar of International Labour Organization (ILO) but it’s not quite clear. Reference is required.
6. Page 5 – Lines 190-191: Since the study was conducted in an Asian country, it would be appropriate to classify the BMI according to Asian cut-offs point.
7. Page 5 – Lines 195-196: Could the classification of “additional diseases” be substituted with the word “co-morbids?” It would be more appropriate. In addition, how were the following clinical conditions confirmed? Was it from patient records? Please mention.
8. Page 4-5 – lines 175-177: The authors set criteria regarding the awareness of DM status, those who were diagnosed by a medical doctor in hospital (self-reported) were considered aware, but those diagnosed with DM during the study were considered unaware? Is this an assumption or a validated classification? Numerous studies have exhibited that awareness of DM is shown among the general population, diabetes patients or non-diabetes patients. Such assumption is contradicting authors effort in studying this variable among diabetes population.

Validity of the findings

Statistical Analysis

1. The authors used Chi-squared tests for univariate analysis. Please report odds ratios (ORs) yielded in the results description. Please remove Chi-square tests values in table 2 and report ORs.
2. Were multi-collinearity between independent variables checked? Were standard errors less than 5 upon conducting logistic regression analysis?
3. The description of results in univariate analysis is incorrect. It should report the ORs as “how likely would this factor be associated with the primary outcome?” Please consult your statistician for further interpretation.
4. Page 9-Lines 252-254: Correlation analysis was not performed, therefore it is inappropriate to say a great correlation between DM status and psychological distress.
5. In table 3, instead of “dashes -”, please state “Ref” to exhibit reference category.
6. What would be the most significant factor associated with psychological distress? This could be explored using the highest Beta value showing level of significance. Please consult your statistician for this evaluation.

Discussion

1. The discussion is a little unfocused, when authors are merely stating comparisons with previous studies. It should be more focused on principle issue, probably the central focus could be discussing the factor most significantly associated with primary outcome measure.
2. It would be interesting to know, what are the gaps of previous studies? What this study adds? How could future studies replicate the findings of this study for more robust investigations?
3. Authors could speculate on how the big health data assessing psychological and behavioural factors through psycho-informatics be used to evaluate these hypotheses in future studies. It would be interesting to know.

Additional comments

The authors conducted a descriptive cross-sectional study that explored the prevalence and correlates of psychological distress among diabetes adults in Jilin province, China. While this is an interesting study, there are several concerns that need attention. The authors should address these concerns thoroughly. If authors have any native English speaking colleagues in his/her institution, I suggest that authors may request advice on English language corrections. You may cite them under acknowledgements.

·

Basic reporting

Minor recommendations
1. Page 3, line 116 'focused on THE LINK BETWEEN psychological disorders and DM,' ....
2. Page 5, line 178, 'We divided education into three levels: primary school and BELOW'...
3. Page 7, line 231, 'psychological distress SIGNIFICANTLY DIFFERED by gender'.....
4. Page 3, line 118, 'The aims of THE study'.
5. Page 3 - Aims of the study: Consider changing aim (3) of the study to an outcome of the study.

Experimental design

No comment

Validity of the findings

1. Greater discussion regarding aim (3) of the study would be good - '(3) to provide proper references for relevant departments to design specific intervention to ...'
2. To ensure that the sample is representative of the province the authors could compare the basic demographics of the sample with the Province population statistics reported by the government department.
3. Since the focus of the paper is DM, consider converting Table 2 so that DM and non DM sample population are compared. i.e univariate analysis of factors associated with DM.

Additional comments

This paper provides a greater understanding of the link between psychological distress and individuals with DM residing in Jilin province China.
Greater emphasis on the policy implications of the findings in the study would be useful.

---

## Round 0.2 · accepted · Accept

The authors well addressed all the concerns raised and suggestions proposed by the reviewers.

·

Basic reporting

No comments

Experimental design

No comments

Validity of the findings

No comments

Additional comments

The authors have addressed all suggested comments and concerns. I am satisfied with the revision.

Good luck on your paper.